# Programmable receptors enable bacterial biosensors to detect pathological biomarkers in clinical samples

Hung-Ju Chang [1], Ana Zúñiga [1], Ismael Conejero[1,2,3], Peter L. Voyvodic[1], Jerome Gracy[1], Elena Fajardo-Ruiz [1], Martin Cohen-Gonsaud[1], Guillaume Cambray [1], Georges-Philippe Pageaux[4], Magdalena Meszaros [4], Lucy Meunier[4] & Jerome Bonnet [1]✉

Bacterial biosensors, or bactosensors, are promising agents for medical and environmental diagnostics. However, the lack of scalable frameworks to systematically program ligand detection limits their applications. Here we show how novel, clinically relevant sensing modalities can be introduced into bactosensors in a modular fashion. To do so, we have leveraged a synthetic receptor platform, termed EMeRALD (Engineered Modularized Receptors Activated via Ligand-induced Dimerization) which supports the modular assembly of sensing modules onto a high-performance, generic signaling scaffold controlling gene expression in *E. coli*. We apply EMeRALD to detect bile salts, a biomarker of liver dysfunction, by repurposing sensing modules from enteropathogenic *Vibrio* species. We improve the sensitivity and lower the limit-of-detection of the sensing module by directed evolution. We then engineer a colorimetric bactosensor detecting pathological bile salt levels in serum from patients having undergone liver transplant, providing an output detectable by the naked-eye. The EMeRALD technology enables functional exploration of natural sensing modules and rapid engineering of synthetic receptors for diagnostics, environmental monitoring, and control of therapeutic microbes.

[1] Centre de Biologie Structurale (CBS), INSERM U1054, CNRS UMR5048, University of Montpellier, Montpellier, France. [2] Neuropsychiatry: Epidemiological and Clinical Research, Inserm Unit 1061, Montpellier, France. [3] Department of Psychiatry, CHU Nimes, University of Montpellier, Montpellier, France. [4] Department of Hepatogastroenterology, Hepatology and Liver Transplantation Unit, Saint Eloi Hospital, University of Montpellier, Montpellier, France. ✉email: jerome.bonnet@inserm.fr

Early disease detection and monitoring of chronic patholo-gies help reduce mortality and improve patients' quality of life[1,2]. In that context, in vitro diagnostic technologies play key roles at different stages of the healthcare chain[3]. However, many diagnostic technologies require heavy equipments, are expensive, and necessitate trained personnel, limiting testing to centralized facilities such as hospitals. Yet, as exemplified by glucose monitoring for diabetes, field-deployable diagnostic devices can tremendously improve patient healthcare, follow-up, and self-reliance[4,5]. Robust, scalable, and cost-effective biosensing technologies for field-deployable diagnostics have thus been under intense research interest over the past decade[6].

Bacteria must sense and respond to myriad chemical and physical signals to survive and reproduce, and are thus ideal candidates for engineering biosensors. Whole-cell biosensors (WCB) are genetically modified living cells that detect molecules of interest, generally using a transcription factor regulated by the ligand of interest, and activating transcription of a reporter gene[7]. While explored since the dawn of genetic engineering[8], recent advances in synthetic biology have improved whole-cell biosensor robustness, sensitivity, signal-to-noise ratio, and signal-processing capabilities, supporting their use in complex media like waste-water and clinical samples[9–11]. With these new capabilities, whole-cell biosensors have the potential to provide miniaturized, field-deployable, diagnostic devices capable of multiplexed detection and sophisticated computation[10,11]. As a self-manufacturing and biodegradable platform, biology provides a cost-effective and environmentally friendly alternative to tradi-tional diagnostic methods. The self-manufacturing nature of biology also offers a unique advantage for low-resource settings and remote, highly constrained conditions, such as those found in space exploration[12]. Despite all these advantages, the scope of application for bacterial biosensors is limited by the difficulty to generate novel sensors detecting biomarkers of interest. Although significant progress has been recently made[13], scalable platforms to rapidly generate new receptors are needed (Fig. 1a).

In order to address this challenge, we recently designed a synthetic receptor platform termed EMeRALD (Engineered Modularized Receptors Activated via Ligand-induced Dimerization)[14] (Fig. 1b). EMeRALD receptors are derived from membrane-bound one-component systems, which are bitopic proteins with a typical architecture of a cytoplasmic DNA-binding domain (DBD), a juxtamembrane linker, a transmem-brane region, and a periplasmic ligand-binding domain (LBD). Direct fusion between the LBD and the DBD provides a simple yet efficient solution to transduce incoming signals into a tran-scriptional output[15,16]. The EMeRALD platform operates in Escherichia coli and uses the DBD from the CadC pH sensor which is inactive in its monomeric state[17]. Ligand-induced dimerization of the LBD triggers dimerization of the cytoplasmic DBD and transcriptional activation[18]. This straightforward mechanism offers the potential to modularize receptor sensing and signaling by domain swapping. We previously built a syn-thetic receptor responding to caffeine by using a nanobody for this ligand as LBD[14]. In this work we attempted to leverage the EMeRALD platform to detect pathological biomarkers. As a pilot application, we aimed to detect bile salts, a biomarker of liver dysfunction.

Liver disease includes dozens of pathologies such as cirrhosis, hepatitis, liver cancer, hepatobiliary problems such as cholangitis, and drug-induced liver injury[19–23]. Liver disease is a global healthcare burden accounting for two million deaths per year, and impacts the quality of life of millions of people worldwide[24]. Liver is the second most common transplanted organ, but only 10% of needs are met. Currently, liver disease diagnostics and monitoring is performed by assessing a panel of biomarkers[25].

However, current methods for in vitro diagnostics are only available in hospitals and testing laboratories, limiting the fre-quency of monitoring for patients. In addition, most biomarkers appear at late disease stages (when significant cellular damage has already occurred) and lack specificity[26]. As liver diseases are chronic, evolutive pathologies, patients would benefit from monitoring devices that enable rapid and simple assessment of liver function with high sensitivity and specificity.

An alternative biomarker of liver dysfunction is the presence of bile salts in serum. Bile salts are key components of bile which are critical for digestion in which they help absorption of fat[27]. Interestingly, serum bile salts have emerged as a general bio-marker of liver disease, and are the gold-standard diagnostic method for pregnancy cholangitis[28,29]. Several studies have also pointed to bile salts as a general biomarker of interest for early cirrhosis, hepatitis, drug-induced liver injury. In addition, bile salts provide a highly specific and dynamic assessment of liver function. For example, serum bile salts are highly correlated with the obstruction state of the bile duct and rapidly decrease when biliary stenting is performed[30]. Furthermore, specific bile salt profiles may be associated with particular liver diseases[31–33]. Bile salts thus represent an ideal and specific biomarker for diagnosis and dynamic monitoring of liver disease. Yet, as for other bio-markers, current detection methods for bile salts based on enzymatic assays[34] are only performed in a centralized fashion and cannot discriminate between different bile salts classes.

In this work we take advantage of the natural capacity of enteropathogenic bacteria to detect bile salts upon arrival into the gut to activate their virulence pathways[35,36]. We build EMeRALD receptor detecting bile salts in E. coli by rewiring bile salt-sensing modules from Vibrio cholerae and Vibrio parahaemolyticus. As the synthetic receptor operates in a surrogate, non-pathogenic host, we perform directed evolution of the sensing module and improve its limit-of-detection (LOD) and sensitivity. Finally, we optimize a colorimetric version of the system to operate in clinical samples. The resulting bactosensor can detect pathological levels of bile salts in serum from patients having undergone liver transplant and provides a signal detectable with the naked eye. Our work highlights the flexibility and modularity of the EMeRALD receptor platform for rapid characterization and engineering of novel sensing capabilities in whole-cell biosensors.

## Results

**Engineering of synthetic bile salt receptors in E. coli using the EMeRALD platform.** Enteropathogenic bacteria such as Vibrio cholerae or Vibrio parahaemolyticus cause acute intestinal infec-tions mediated by toxin secretion[37]. These pathogens use bile salts as an intestinal location signal to activate their virulence path-ways. Bile salt sensing is mainly under the control of inner membrane sensor/cofactor couples TcpP–TcpH for V. cholerae[35] and VtrA–VtrC for V. parahaemolyticus[38]. As using pathogens as biosensors would involve significant host-specific regulation correction[38,39] and biosafety containment issues, an alternative strategy is to rewire pathogen-sensing modules of interest into a modular receptor platform operating in a surrogate host (Fig. 1a).

To engineer an EMeRALD bile salt receptor in E. coli, we fused the V. cholerae TcpP bile-salt-sensing module and its transmem-brane region (TM) to the DNA-binding domain of CadC (Fig. 1c). As a reporter, we placed superfolder green fluorescent protein (sfGFP)[40] under the control of the CadC target promoter, pCadBA. We also expressed the cofactor protein TcpH, previously described to protect TcpP from proteolysis by the V. cholerae RseP protease once dimerized in response to bile salts[41]. We first confirmed using inducible gene expression systems that co-expression of the TcpH cofactor was necessary for TcpP

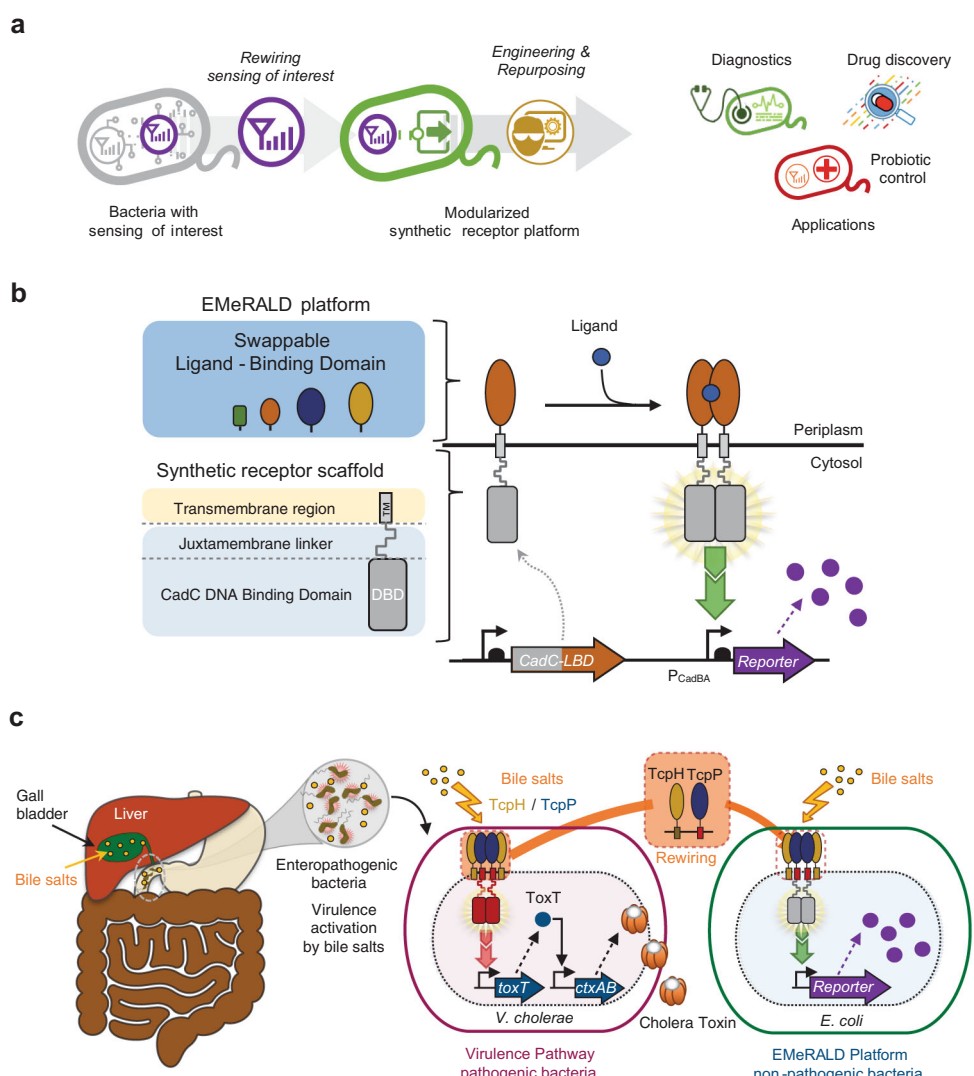

**Fig. 1 Design principles and architecture of EMeRALD-based bacterial sensors. a** General strategy of rewiring bacterial sensing modules into surrogate hosts using a synthetic receptor platform. **b** Architecture and functional components of EMeRALD platform. The EMeRALD platform is composed of swappable ligand-binding domains (LBD) that can be plugged into a synthetic receptor scaffold consisting of the DNA-binding domain (DBD) of the CadC protein, a juxtamembrane (JM) linker, and a transmembrane region. The resulting synthetic transmembrane receptor is activated via ligand-induced dimerization and triggers reporter gene expression. **c** Rewiring bile salt sensing into *E. coli* using the EMeRALD platform. Enteropathogenic bacteria detect intestinal bile salts as host environmental cues for activating their virulence pathway. We plugged the *V. cholerae* bile salt receptor TcpP, and its cofactor protein TcpH, into the EMeRALD platform operating in the surrogate host *E. coli* to build a synthetic bile salt receptor controlling expression of a reporter gene.

function, using the primary bile salt taurocholic acid (TCA) as a ligand (Supplementary Figs. 1–3). These data suggest that the RseP homolog present in *E. coli* (UniProt:P0AEH1) can degrade TcpP when not bound by TcpH. We also found that the relative expression level of CadC-TcpP and TcpH were critical parameters affecting system performance (Supplementary Fig. 3).

We then placed both proteins under the control of constitutive promoters[42]. We used the strong constitutive promoter P5 for TcpH and three different constitutive promoters of increasing strengths, P9, P10, and P14, for CadC-TcpP (Fig. 2a) and tested their response to TCA[41] (Fig. 2b and Supplementary Figs. 4–5) (Promoter strength: P5 > P14 > P10 > P9 (ref. [42])). We found that the P9-CadC-TcpP variant had the lowest LOD, highest dynamic range, and highest signal strength. These data confirm that the stoichiometry between CadC-TcpP and TcpH is a key parameter influencing receptor performance.

We then assessed the versatility of the EMeRALD platform by connecting the VtrA/VtrC sensor system from *V. parahaemolyticus*[36] (Fig. 2c). We built a dual-expression system consisting of P9-CadC-VtrA and P5-VtrC, and tested its response to its canonical ligand taurodeoxycholic acid (TDCA). We found that the VtrA/VtrC EMeRALD system was functional with a slightly higher LOD, similar dynamic range and signal strength than the TcpP/TcpH EMeRALD system (Fig. 2d and Supplementary Fig. 6). These results highlight the modularity and scalability of the EMeRALD platform, which supports the connection of different sensing modules to the receptor scaffold.

**Synthetic bile salt receptors exhibit different specificity profiles.** We then assessed the specificity profile of the synthetic bile salt receptors. Bile salts are classified in two categories: primary

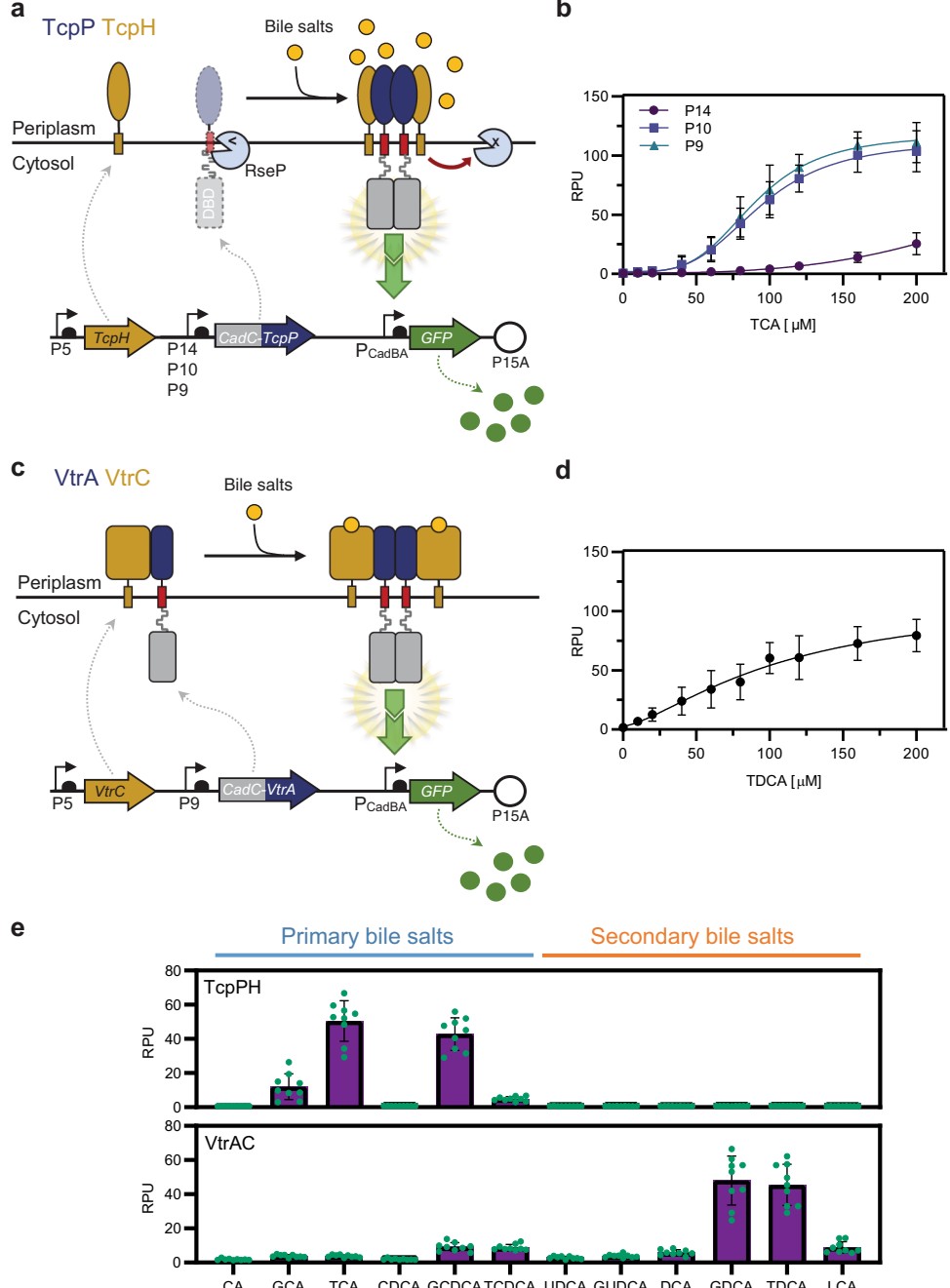

**Fig. 2 Design, implementation, and characterization of synthetic bile salt sensors. a** Overview of the TcpPH-EMeRALD system. The CadC DNA-binding domain (DBD) is fused to the transmembrane and periplasmic domains of TcpP. Three constitutive promoters (P14, P10, and P9) were tested to tune the transcription level of CadC-TcpP. Transcription of the TcpH cofactor is under the control of the constitutive promoter P5. In the absence of bile salts, CadC-TcpP is probably degraded by an endogenous *E. coli* homolog of the *V. cholerae* protease RseP. In the presence of bile salts, CadC-TcpP dimerizes and forms a stable complex with TcpH that protects it from proteolysis. The CadC-TcpP dimer then activates downstream expression of the GFP reporter. **b** Transfer function of TcpPH-EMeRALD receptors controlled by different promoters in response to increasing concentrations of the bile salt taurocholic acid (TCA). **c** Overview of the VtrAC-EMeRALD system. The CadC DBD is fused to the transmembrane and periplasmic domains of VtrA. CadC-VtrA and VtrC are under the control of the P9 and P5 promoters, respectively. Bile salts binding to VtrA/VtrC heterodimeric complexes promote oligomerization of CadC-VtrA and activate downstream expression of the GFP reporter. **d** Transfer function of VtrAC-EMeRALD receptor to increasing concentrations of the bile salt taurodeoxycholic acid (TDCA). **e** Bile salt specificity profiles for TcpPH-EMeRALD and VtrAC-EMeRALD systems. The full names and molecular structure of the different bile salts are listed in Supplementary Fig. 1. The curve graphs (**b, d**) correspond to the mean value of three replicates performed in triplicate on three different days (*n* = 3 biologically independent samples). The bar graph (**e**) corresponds to the mean value of three replicates performed in triplicate on three different days (*n* = 3 biologically independent samples). Green dots correspond to the values for all replicates. Error bars: ±SD. RPU reference promoter units. Cells growing in exponential phase were incubated with bile salts for 4 h before flow cytometry measurement.

bile salts (including TCA) are produced by the liver while secondary bile salts arise from modification of primary bile salts by gut microbiome metabolism. Primary bile salts are upregulated in serum and urine of patients with liver disease[31–33]. Previously identified virulence activating factors for *V. cholerae* include TCA, glycocholate, and cholic acid[41]. We measured the response of the bactosensor to a panel of 12 different bile salts, including both primary and secondary types (Fig. 2e and Supplementary Figs. 7 and 8). Interestingly, while no sensing module was specific for a single bile salt species, the CadC-TcpP system was highly specific for primary conjugated bile salts (especially TCA and GCDCA), and did not respond to secondary bile salts. On the other hand, the CadC-VtrA system had a larger spectrum of bile salt specificity, mainly responding to secondary conjugated bile salts GDCA and TDCA. Due to the link between primary bile salts and liver diseases, we selected the TcpP/TcpH system to develop a bile salt bactosensor for medical diagnosis.

**Directed evolution of TcpP-sensing module for improving LOD and higher sensitivity**. Sensor sensitivity and LOD are key parameters for biosensors applications. We aimed to identify key residues determining the sensitivity of the TcpP-sensing module, and targeted those to improve synthetic receptor sensitivity and LOD. To do so, we coupled comprehensive mutagenesis with functional screening and next-generation sequencing (NGS), an approach that supports the identification of functional variants together with the sequence determinants within local structural motifs[43–45]. This strategy has also been used to engineer orthogonal two-component systems[46].

Transition from intramolecular to intermolecular disulfide bonds between TcpP monomers is a key determinant of TcpP response to bile salts and is mediated by two cysteine residues, Cys207 and Cys218 (ref. [41]). By performing multiple sequence alignments of different TcpP bacterial homologs (Supplementary Fig. 9), we found a significant conservation of the amino acids flanked by these two cysteines (Fig. 3a). Secondary structure prediction and ab initio 3D prediction using the Rosetta modeling suite (Fig. 3b and Supplementary Fig. 10) suggested that each cysteine was located in rigid beta sheets separated by a flexible loop region between Asn210 and Gln213. This loop propensity to form a turn would allow the two beta sheets and the cysteines to come in close proximity and form an intramolecular disulfide bond. We hypothesized that the flexibility of the turn region between Cys207 and Cys218 was a key parameter controlling the transition rate between the two states, and that altering its amino acid composition could change the system's sensitivity to bile salts.

We thus built a comprehensive mutational library (NNK x 4, theoretical library complexity $\cong 1.05 \times 10^6$ variants, see "Methods" for details) targeting the NYEQ residues inside the turn, and cloned it into the plasmid constitutively expressing CadC-TcpP and producing GFP in response to bile salts (Fig. 3c). The resulting library was induced with TCA, and GFP-positive variants were isolated by fluorescence-activated cell sorting. We performed three rounds of enrichment (200 μM of TCA as ligand in first and second rounds of selection, and 20 μM for the third round) and observed an increasing fraction of the cell population responding to different ligand concentrations (20–80 μM) (Fig. 3d). We collected, cultured, and sequenced single variants and tested their response to TCA (Fig. 3e). We found that comprehensive mutagenesis of residues Asn210 to Gln213 could alter the limit-of-detection, the sensitivity, and the fold activation of our biosensor. The 3.3-fold difference in LOD between V18 and V22 ($EC_{50}$ from 28.3 to 92.5 μM, Table 1) indicated the broad range of sensitivity engineering obtained by mutating the loop

region of the TcpP-sensing module. Further kinetic analysis revealed that the sequence variation of this loop region changes reaction speed and system interaction of bile salts with the synthetic receptor, and variant V18 has 13-fold increase in ligand affinity and faster response at low ligand concentration compared to wild type TcpP (Supplementary Fig. 11).

To better understand the sequence features influencing the response of TcpP to bile salts, we sequenced the whole pool of enriched variants by NGS. Surprisingly, the sequence features of functional variants were different from those expected from natural TcpP homologs (Fig. 3f and Supplementary Fig. 12). First, and in contrast with wild type TcpP homologs, we observed a strong depletion of long-chain, negatively charged amino acids (Asp and Glu) along with long-chain polar amino acids (Asn and Gln) at position 211. Lysine at position 211 also appeared to be depleted in functional variants (despite being commonly found at this position in other TcpP homologous proteins). Second, amino acids with bulky aromatic side chain such as Phe and Tyr, and hydrophobic side chain such as Leu were highly conserved in selected functional variants, strongly indicating the important role of hydrophobic residues at position 211 in the C-terminal loop region for the function of *V. cholerae* TcpP. We chose the best engineered variant, termed TcpP18, for further development of a clinical bactosensor.

**Development of a colorimetric version of the bactosensor**. Colorimetric assay provides a simple and intuitive method for simple and direct estimation of test results by the naked eye. In addition, colorimetric assays support straightforward development of quantitative assays using smartphone-based platforms for POC or home-based diagnosis[47]. We used TcpP18 coupled with the reporter beta-galactosidase LacZ (termed TcpP18–LacZ) and its substrate chlorophenol red-β-D-galactopyranoside (CPRG) to provide a colorimetric output[48] (Fig. 4a, see "Methods" for details). Similarly to the biosensor equipped with a GFP output, the bile salt specificity profile of the TcpP18–LacZ system was slightly shifted from TCA to GCDCA (Fig. 4b and Supplementary Fig. 13). We thus evaluated the LOD and signal output threshold of TcpP18–LacZ in response to increasing concentrations of GCDCA. We also explored the influence of varying cell density and incubation time (Fig. 4c and Supplementary Figs. 14 and 15). By adjusting cell density or incubation time we improved the dynamic and operating ranges of TcpP18–LacZ (Fig. 4c and Supplementary Figs. 14 and 15). After optimization, the Tcp18–LacZ demonstrated linear response to GCDCA within the concentration from 0 to 40 μM in 1 h (Supplementary Fig. 15). This allowed us to tune the threshold activation level to match various clinical levels associated with specific liver-related medical conditions.

**Bactosensor-mediated detection of elevated bile salts levels in serum from patients with liver transplant**. We then prototyped our bactosensor for the detection of bile salts in clinical conditions. To do so, we tested the sensor on samples from patients having undergone liver transplantation. After liver transplant, the main complications are bile ducts stenosis and acute cellular rejection (ACR). In order to detect these complications at an early stage, liver tests are performed regularly. Serum bile salt concentration has been shown to be a good indicator for the assessment of liver dysfunction after liver transplantation[49]. A field-deployable method for bile salt assessment would greatly improve the monitoring of these patients, ultimately allowing fine grained monitoring performed at-home by the patients themselves.

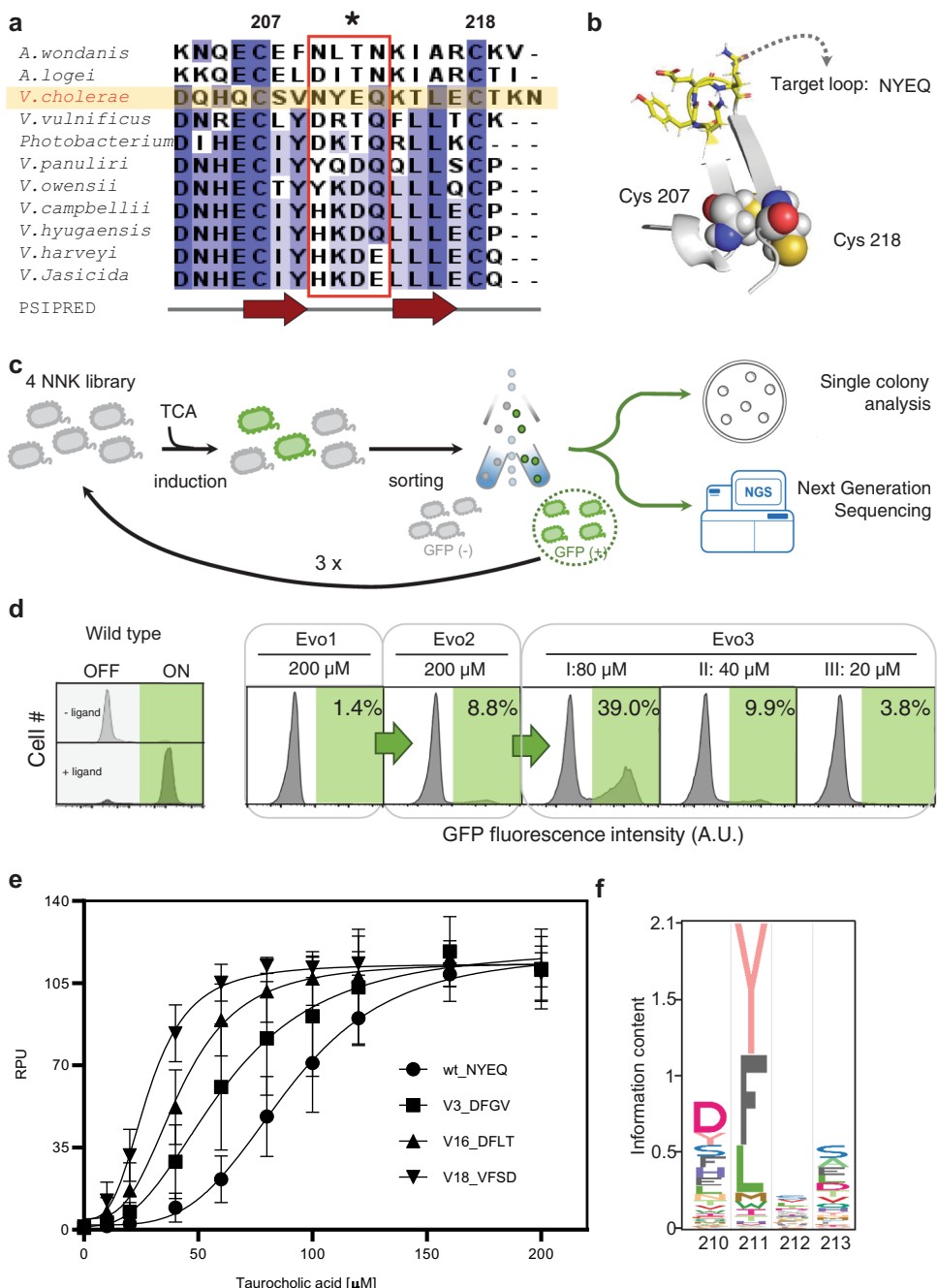

**Fig. 3 Directed evolution of TcpP for improving LOD and sensitivity. a** Multiple sequence alignment of the C-terminal periplasmic domains from TcpP homologous proteins. Colors indicate sequence identities. Secondary structures prediction by PSIPRED is shown below. The red line box indicates the region of interest for mutational scanning. **b** Ab initio modeling of the C-terminal region of the TcpP-sensing module. The two cysteine residues are shown as spheres. The amino acid residues NYEQ in the loop region of interest are labeled as sticks. **c** Schematic diagram of the screening procedure to obtain functional TcpP loop variants. The cell library was submitted to several rounds of sorting based on fluorescent signal output produced in response to bile salts. Individual clones were then isolated and characterized while the sequence of the mutated loop for the whole pool of isolated variants was analyzed by NGS. **d** Screening conditions and results of TcpP directed evolution. The gate used for cell sorting is colored in green. The x-axis indicates the fluorescence intensity in arbitrary units (AU). The gate for fluorescence-activated cell sorting is determined using the fluorescence distribution of TcpPH-EMeRALD sensor with or without the 200 μM TCA (left panel). **e** Transfer function of TcpP functional variants with improved sensitivity in response to increasing concentrations of TCA. Amino acid sequences of the loop regions are indicated. The curve graphs correspond to the mean value of three replicates performed in triplicate on three different days ($n = 3$ biologically independent samples). Error bars: ±SD. RPU reference promoter units. **f** Sequence logos of the flexible loop region from selected TcpP functional variants derived from NGS data.

We tested our bactosensor in 21 serum samples from liver transplantation patients (Fig. 5a and Supplementary Fig. 16). The patients were followed at the Montpellier hospital after their liver transplant, most of them having been performed in the last 2 years (see Supplementary Tables S2 and S3). These patients had received a liver transplant for end-stage liver disease as a result of alcoholic related liver disease or non-alcoholic fatty liver disease, chronic cholangitis, or liver cancer. A complete hepatic check-up

**Table 1 Functional analysis of selected TcpP functional variants.**

| # Variant | Sequence | EC$_{50}$ (μM) | Response EC$_{50}$ (RPU) | Max fold change |
|-----------|----------|--------------|--------------------------|-----------------|
| V22 | YYVL | 92.531 | 56.19 | 69.06 |
| TcpPwt | NYEQ | 89.405 | 56.5 | 164.74 |
| V7 | WYVH | 86.029 | 53.37 | 51.31 |
| V78 | FYES | 84.668 | 59.62 | 69.06 |
| V11 | YYIV | 82.167 | 53.83 | 31.76 |
| V14 | TFLA | 81.636 | 62.97 | 222.67 |
| V3 | DFGV | 61.48 | 61.18 | 142.21 |
| V19 | FFKA | 59.186 | 61.29 | 127.75 |
| V16 | DFLT | 42.058 | 59.34 | 75.73 |
| V18 | VFSD | 28.344 | 58.99 | 84.92 |

was performed, and serum bile salts were also measured using an enzymatic assay (Supplementary Table S4 for clinical data). We noticed that our bactosensor exhibited a negative response to some clinical serum samples. Therefore, we tried to minimize the potential interferences by sample dilution and compared the readouts with serum bile salt concentration measured by enzymatic assay to determine the proper dilution factor (Supplementary Fig. 16). We found that patients who had a high potential of ACR after liver transplantation (serum bile acid >37 μM)[50] had significant and visible colorimetric signal changes in bactosensor assays (Supplementary Fig. 17). Three patients in particular raised our attention: patients #5, 10, and 13. These three patients had elevated serum bile salts concentration. Two of them (5 and 10) presented abnormalities in their hepatic enzymatic values (Aspartate Aminotransferase (ASAT), Alanine Aminotransferase (ALAT), gamma-glutamyl transferase (GGT), Alkaline Phosphatase (ALP), and bilirubin). For these patients, the bile salt bactosensor produced the strongest colorimetric change easily detectable with the naked eye (Fig. 5b). These results indicate that our bactosensor is able to provide a simple, reliable, and cost-effective method for monitoring patient condition after liver transplantation.

## Discussion

Microbes detect and process myriad environmental signals, providing a vast sensing repertoire for engineering biosensors usable for several applications. However, the intricacy of signaling networks and our limited understanding of their biochemical properties restrain their direct use. Here we presented a general strategy to rewire sensing modules of interest into a well-characterized synthetic receptor platform, EMeRALD, enabling their fine tuning and repurposing for biomedical applications. In the future, these receptors could be coupled with genetic circuits performing multiplexing logic, memory, and signal amplification to engineer even more sophisticated whole-cell biosensors[10,51].

We were able to connect two different bile salt-sensing modules having different specificity profiles, the TcpP/H and VtrA/C systems, which respond mostly to primary and secondary bile salts, respectively. Importantly, our capacity to engineer synthetic bile salt signaling using only these protein domains demonstrates that these modules are the only essential components required for bile salt sensing in their natural host. By performing directed evolution of TcpP, we improved the sensitivity and decreased the LOD of the sensor. In addition, we discovered previously unknown amino acid sequence features influencing TcpP function and potentially relevant for *V. cholerae* virulence. For instance, we found a stringent functional requirement for amino acids with a nonpolar aromatic side chain at position 211. This requirement indicates rigorous steric interactions located in the

C-terminal loop region of *V. cholerae* TcpP. The exact role of Tyr211 is still unclear, but this residue may affect TcpP–TcpP dimer formation or TcpP–TcpH interaction. While the unique presence of Tyr at the 211 position in *V. cholerae* was visible on multiple sequence alignments, the link between this residue and TcpP function could not be inferred by this approach.

This discovery exemplifies how our synthetic receptor platform offers a powerful strategy to interrogate the sequence–function relationship of bacterial sensing modules in a massively parallel fashion. Such studies in the natural pathogenic host would have been tedious because of complex pathway regulations and safety issues would have limited the final bactosensors to a few expert groups. In contrast, our platform provides a straightforward and scalable strategy to study pathogen signaling in surrogate hosts, usable by the larger scientific community. The EmeRALD system coupled with the deep-mutational scanning, methods to navigate protein sequence space[52–55], designer libraries, and Flow-Seq analysis[45] can serve as a general platform for studying bacterial sensing modules, engineering their ligand specificity and their response properties. In particular, our system is ideal for understanding transmembrane one-component signaling, which mechanism and regulation are currently underexplored[16]. On the same line, EMeRALD receptors could be applied to the discovery of inhibitors of virulence pathways through screening of chemical or natural substances libraries. Finally, the bile salt-sensing system could be applied to control the activity of engineered probiotics upon arrival in the gut, as recently proposed in *Bacteroides thetaiotomicron*[56].

Here we show a pilot application of the EMeRALD technology to the field of medical diagnostics by detecting abnormal bile salt levels in patient samples. As bile salts dysregulation and gut dysbiosis are critical in the pathogenesis of liver diseases or gastrointestinal cancer[57–62], there is an urgent need for POC assays for bile salt monitoring[63]. Our bactosensor results correlate well with hospital tests and produce a very strong signal detectable with the naked eye for the three most critical patients, using very small sample volumes, demonstrating the potential of our technology for future POC estimation of liver dysfunction.

Yet, several improvements are needed to translate our platform into a POC or home diagnostics system. First, given the size of our patient cohort and its diversity, further studies are required to fully validate our approach in a clinical context. Second, while our work demonstrates the clinical applicability of our platform–our assay by itself is still not compatible with POC operation. In particular, it still includes several liquid handling and incubation steps, such as cell growth, spinning, and lysis. Lyophilizing the bactosensors once the first preparations steps have been done would provide a simple test format. Several examples have shown the possibility to lyophilize bacteria and maintain their function, even on paper[64]. The coupling of whole-cell biosensors with microfluidics devices into a lab-on-chip apparatus has also been extensively documented[51,65] and could be applied to our system. Finally, the use of pigment reporters or the implementation of autolysis systems could suppress the additional step to lyse cells[66–68].

Regarding the biological sample itself, serum preparation from blood requires additional manipulations. While those can be performed in POC using low-tech devices such as the paperfuge, detecting bile salts in more readily accessible physiological fluid might be advantageous. Urine offers such an opportunity, with the added benefits of a non-invasive collection process. We indeed validated the function of our bactosensor to detect exogenously bile salts in urine samples (Supplementary Fig. 19). However, detecting endogenous bile salts in patients' urines, even with high bile salt levels, proved unreliable. We attribute this issue to the fact that most bile salts excreted in urines are sulfonated

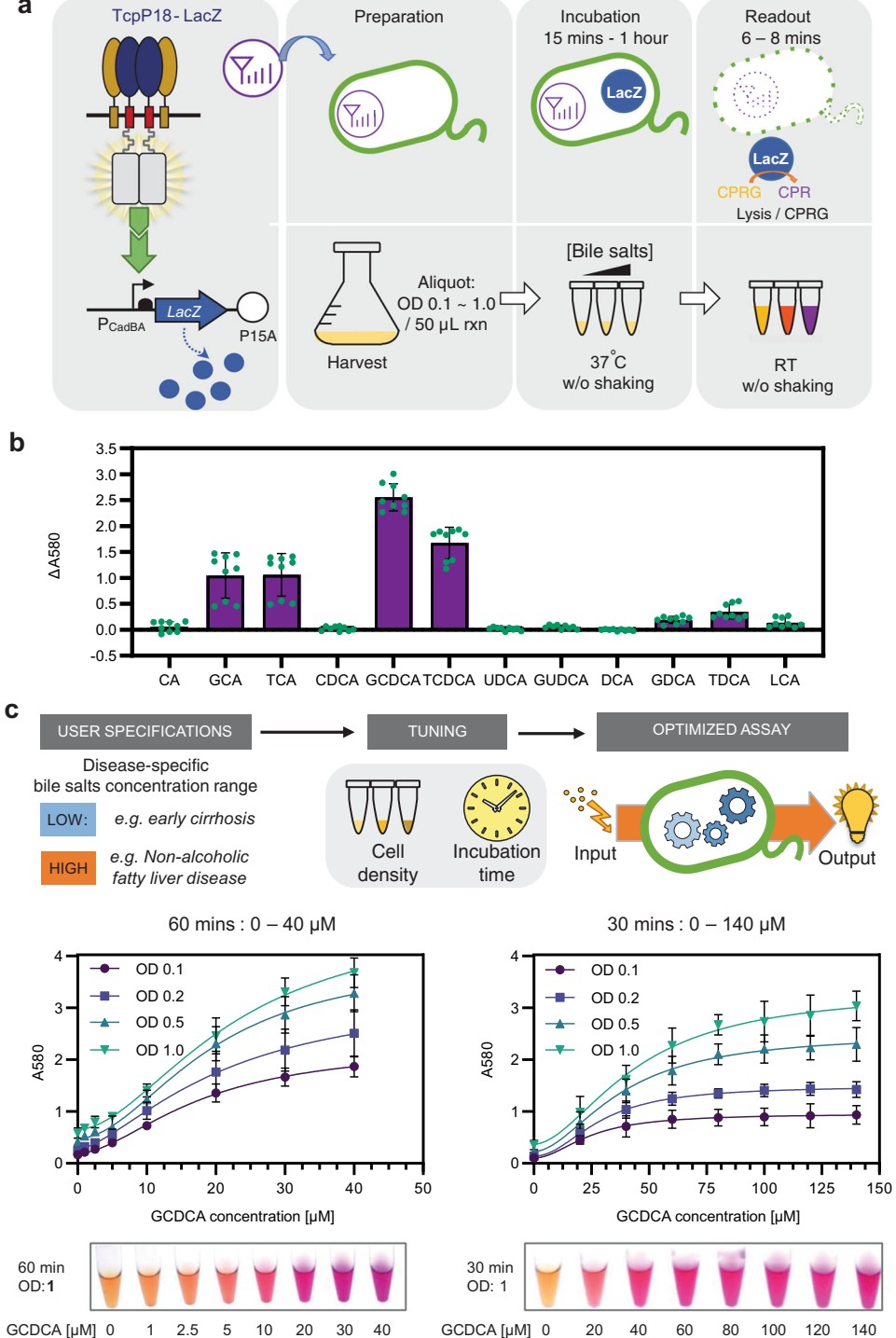

**Fig. 4 Colorimetric assay for bactosensor-mediated bile salts detection. a** Schematic diagram for the design and operation procedure of the TcpP18–LacZ system for bile salt detection. The TcpP18–LacZ uses the TcpP loop variant V18 as sensing module for lower LOD and higher sensitivity and LacZ as a colorimetric reporter, with CPRG as a substrate which is converted to chlorophenol red turning the reaction from yellow to purple. **b** Bile salt specificity profile of the TcpP variant V18 characterized using TcpP18–LacZ sensor. Response of TcpP18–LacZ was quantified as ΔA580 (the difference in absorbance at 580 nm ($A_{580}$) with or without ligand bile salts). The bar graph corresponds to the mean value of three replicates performed in triplicate on three different days ($n = 3$ biologically independent samples). Error bars: ±SD. Cells growing in the exponential phase were incubated with bile salts for 4 h before flow cytometry analysis. **c** Optimization and quantification of TcpP18–LacZ response to the bile salt glycodeoxycholic acid (GCDCA). The LOD and response dynamic range of TcpP18–LacZ were fine-tuned by varying the cell concentration and incubation time. Data points correspond to the mean value of three replicates performed in triplicate on three different days ($n = 3$ biologically independent samples). Error bars: ±SD. See "Methods", main text, and SI for details.

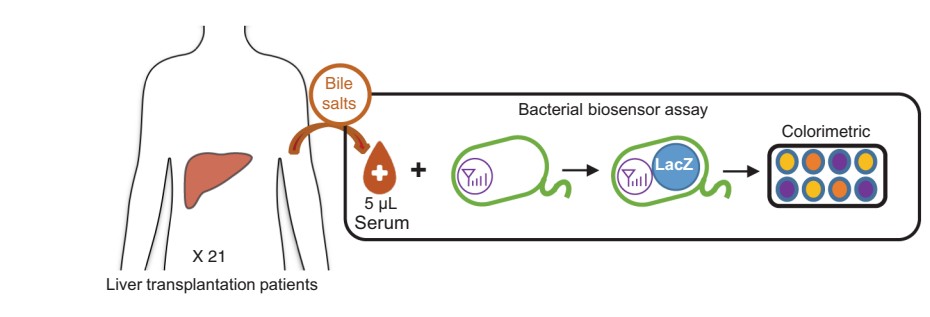

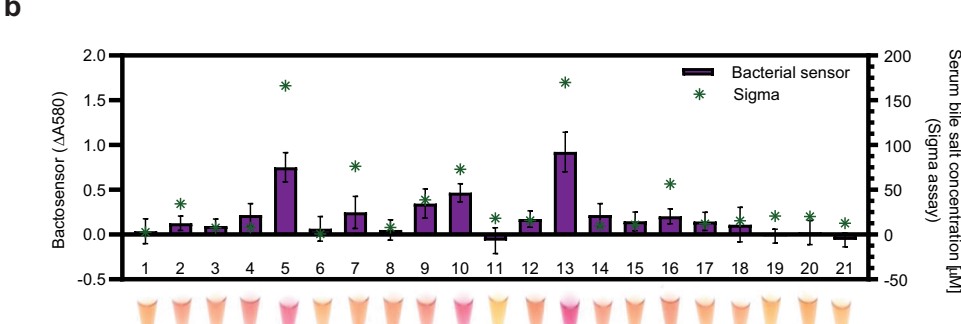

**Fig. 5 Bactosensor-based pathological bile salt detection in clinical samples. a** Serum samples from 21 liver transplantation patients being followed at a Montpellier hospital were analyzed using a bactosensor equipped with the TcpP18–LacZ sensor as described in Fig. 4a, with a 10-fold dilution and 2 h incubation time (Supplementary Fig. 14). **b** Comparing the analysis result of 21 clinical serum samples between TcpP18–LacZ and a bile salt assay kit. The response of TcpP18–LacZ is shown in purple bars, left axis. The serum total bile salt concentration measured by a bile salt enzymatic assay kit is labeled in green asterisks, right axis. The bar graph corresponds to the mean value of three replicates performed in triplicate on three different days (n = 3 biologically independent samples). Error bars: ±SD. See Supplementary Fig. 18 for plots with all replicates represented.

after they pass through the kidney[69], and that our biosensor seems unable to detect sulfonated versions of bile salts (Supplementary Fig. 20). This issue could be solved by treating samples or expressing in bacteria a bile acid sulfate sulfatase[70], or engineering the TcpP or VtrA modules to detect sulfonated bile salts. The specificity shift observed with TcpP-18 compared to the wild type TcpP suggests that such engineering is feasible.

We observed that our bactosensor had relatively low or negative responses to several serum samples (Fig. 5b and Supplementary Fig. 16). This behavior may be due to differential matrix effects arising in samples from different patients. Large-scale analysis of clinical samples chemical composition coupled with close monitoring of the diet and medical treatments for each patient could help identify the causes of this effect.

Finally, our sensing platform based on genetically engineered organisms will require containment systems before being deployed[71,72], together with open ethical and societal debate[73]. Addressing most of these challenges entails repurposing existing systems and frameworks, together with solving engineering puzzles that are within current trends of synthetic biology and biosensor research.

Here we have shown that the EMeRALD platform, which can accommodate natural or synthetic sensing modules, provides a versatile and scalable solution to develop new sensing modalities in bacteria, with the potential to help address practical challenges. We anticipate that EMeRALD receptors will be repurposed for other applications in medical diagnostics, bacterial therapeutics, and environmental monitoring, supporting the transition of bacterial biosensors towards a wide range of real-world applications.

## Methods

**Plasmids and strains**. Genetic parts of constructs used in this study are provided in Supporting Information. All constructs were cloned into plasmid J64100_p15A

with p15a origin of replication and chloramphenicol resistance gene by isothermal Gibson assembly. All experiments were performed using *E. coli* strain NEB10β (New England Biolabs). Plasmids and materials will be made available through Addgene.

**Functional characterization of synthetic bile salt receptors with sfGFP fluorescence outputs**. For the experiments of constructs with constitutive promoters (Figs. 2b, d, e, and 3e), plasmids encoding different constructs were transformed into chemically competent *E. coli* NEB10β (New England Biolabs), plated on LB agar plates supplemented with 25 μg/mL chloramphenicol and incubated at 37 °C overnight. For each measurement, three fresh colonies were picked and inoculated into 5 mL of LB/chloramphenicol and grown at 37 °C with vigorous shaking for 16–18 h. In the next day, the cultures were diluted 1:100 into 1 mL of LB/chloramphenicol medium with different concentration of bile salts in 96 deep-well plates (Greiner bio-one), incubated at 37 °C with vigorous shaking for further 4 h and analyzed by flow cytometry. All experiments were performed at least three times in triplicate on three different days. For bile salt specificity profiles of TcpPH-EMeRALD and VtrAC-EMeRALD systems (Fig. 2e), experiments were performed with the same protocol, and using each bile salt at a 80 μM concentration. For the experiments of constructs with inducible promoters (Supplementary Fig. 2), the overnight cultures were diluted 1:100 into 1 mL of LB/chloramphenicol medium with different concentrations of IPTG, 1.5 mM of benzoic acid, and different concentrations of bile salts in 96 deep-well plates. And then follow the same procedures as constructs with constitutive promoters. All chemicals used in this research were purchased from Sigma-Aldrich.

**Calculation of relative promoter units (RPUs)**. Fluorescence intensity measurements among different experiments were converted into RPUs by normalizing them according to the fluorescence intensity of an *E. coli* strain NEB10β containing a reference construct and grown in parallel for each experiment. We used the constitutive promoter J23101 and RBS_B0032 as our in vivo reference standard and placed superfolder GFP as a reporter gene in plasmid pSB4K5. We quantified the geometric mean of fluorescence intensity (MFI) of the flow cytometry data and calculated RPUs according to the following equation:

$$RPU = (MFI_{sample})/(MFI_{reference\ promoter}) \qquad (1)$$

**Flow cytometry analysis**. Flow cytometry was performed using an Attune NxT cytometer coupled with high-throughput autosampler (Thermo Fisher Scientific)

and Attune NxT™ Version 2.7 Software. In all, 30,000 cells were collected for each data point. Experiment on Attune NxT were performed in 96-well plates with setting; FSC: 200 V, SSC: 380 V, green intensity BL1: 440 V (488 nm laser and a 510/10 nm filter). Flow cytometry data were analyzed using FlowJo 10.0.8r1 (Treestar Inc., Ashland, USA). All raw data values are listed in Supplementary Information. The gating strategy is shown in Supplementary Fig. 21.

**4 x NNK library construction**. Using P9-CadC-TcpP plasmid as a template, the insert with 4 x NNK library was amplified by Phusion Flash High-Fidelity PCR Master Mix (Thermo Fisher Scientific) with the primers I5 and I3; the vector was amplified with the primers V5 and V3 (primer sequence details are listed in Supplementary Table 9). The purified fragments were first digested with BsaI-HFv2 (New England Biolabs), and then ligated by T4 ligase at 4 °C overnight. Five micrograms of ligation product was transformed into *E.coli* strain NEB10β with electroporation.

**Cell sorting**. Cell sorting was performed using a S3 cell sorter (Bio-rad) with Bio-rad S3 software Prosort version 1.6. In total, 100,000 cells under different induction conditions (as shown in Fig. 3) were gated and collected in SOC medium at each round of sorting. Experiments on Bio-rad S3 cell sorter were performed with setting; FSC: 400 V, SSC: 284 V, green intensity FL1: 680 V (488 nm laser and a 510/10 nm filter). The sorted cells were further inoculated in 10 mL of LB/chloramphenicol medium at 37 °C with vigorous shaking for a further 16–18 h. The cell cultures positively selected cells were then applied to the next round of selection. For the first two rounds, the loop variant library was induced with 200 μM of TCA. In the third round of evolution, cells were further induced at different concentrations of TCA to isolate more sensitive variants. For each round of screening, cells from an overnight culture were diluted 1:100 in LB with or without TCA and grown for 16 h at 37 °C before being sorted. The gating strategy is shown in Fig. 3d and Supplementary Fig. 22.

**Rosetta modeling**. Ab initio structural modeling. Structural models of the TcpP C-term segment (residues 182 to 211) were generated using the ROSETTA ab initio 3D prediction protocol[74] with 1D sequence and 2D predicted secondary structure as input data.

**Next-generation sequencing**. Cells sorted from the third round of evolution were further induced with 320 μM of TCA to collect the fully activated TcpP variants. Plasmids extracted from sorted cells were further amplified (primer sequence details are listed in Supplementary Table 9) and added UMI barcodes with first round primers (see Supplementary Materials for sequence details). After PCR clean up, the first round PCR products were further amplified by second round PCR primers with NGS index. The PCR products were further purified with AMPure XP paramagnetic beads to remove the contaminants. Samples were sequenced on Illumina PE250 platform at Novogen (Hong Kong, China) with paired-end reads of 250 bases.

**NGS data processing**. The datasets generated and analyzed during the current study are available in the NCBI SRA repository (BioProject ID: PRJNA714981; https://www.ncbi.nlm.nih.gov/bioproject/714981). Sequence counts were converted into amino acid sequences and listed in Supplementary Data 1. The python scripts for NGS data processing are available from Github Page: https://github.com/hungjuchang/NGS-Sequence-Counts_PWM_PSSM-calculator.git. The resulting data were used for sequence logo preparation by R package Logolas version 1.3.1 (ref. [75]).

**Time-course measurements**. For Supplementary Fig. 11, bacteria cells containing different CadC-TcpP promoter variants (with P5-TcpH) were streaked on LB plus 1.5% agar plates and grown overnight at 37 °C. Threes different colonies were inoculated into 1 mL of LB plus antibiotics in a 2 mL 96 deep-well plates (Thermo Fisher Scientific, 278606) sealed with AeraSeal film (Sigma-Aldrich, A9224-50EA) and incubated at 37 °C for 16 h with shaking (300 r.p.m.) and 80% of humidity in a Kuhner LT-X (Lab-Therm) incubator. After overnight growth the cells were diluted 1:250 into 200 μL 96-well plates with M9 minimal medium with 0.4% glycerol plus different TCA concentration. Measurements were done on a Cytation3 microplate reader (Biotek Instruments, Inc.). Absorbance at 600 nm and GFP (excitation 485 nm, emission 528 nm, gain 80) were measured every 10 min for 4 h (linear range of growth). Data were collected using a Gen5 Microplate Reader and Imager Software version 3.03. Raw data were processed by subtracting autofluorescence and normalizing by OD values. Michaelis–Menten enzyme kinetics was evaluated by using GraphPad Prism (version 8.0.2) nonlinear regression model.

**Colorimetric assay of bactosensors for bile salt detection**. Overnight cultures of TcpP–LacZ were diluted in 1/250-fold into 100 mL of LB with 25 μg/mL chloramphenicol, incubated at 37 °C for 4 h to reach an $OD_{600}$ of around 0.4. The cells were put on ice for 30 min and then spun down at 4 °C and 2200g for 10 min. The cell pellets were resuspended with LB medium (without antibiotics) at 1.1× of the final OD. Five microliters of different bile salts (prepared in LB medium) was added into 45 μL of resuspended cells in 96-well plates. The plate was incubated at 37 °C for 10–60 min without shaking. After incubation, 50 μL of B-PER solution (Thermo Fisher) with 800 μM CPRG was added into the cell culture directly. Due to the low permeability of substrate CPRG, it would take more than 2 h to have slight visual colorimetric change, and the difference between with or without ligand is much less than the lysis method. The mixture was further incubated for 6 min at room temperature and then 50 μL of 1 M sodium carbonate was added to stop the reaction. The absorbance at 580 nm was measured with a plate reader (Biotek; Cytation3). For the clinical sample assay, serum samples were heat-inactivated by incubation in a 56 °C water bath for 30 min. Five microliters of serum samples were added into 45 μL of resuspended cells in a 96-well plate and incubated at 37 °C without shaking for 2 h before lysis.

**Statistical analysis**. The signal outputs with significant differences (compared with the signal without ligand) verified by unpaired two-tailed Student's $t$-test are marked by asterisk ($*p < 0.05$, $**p < 0.01$, $***p < 0.001$, and $****p < 0.0001$, respectively). Dose–response curves were fitted using a nonlinear regression model with Hill Slope (four-parameter dose–response curve). All data analysis was performed on GraphPad Prism (Version 8.0.2).

**Patient samples**. Serum samples were consecutively collected from hospitalized or ambulatory liver transplant patients at the Hepatology and Liver Transplantation Unit, Hôpital Saint-Eloi in Montpellier (France) in the month of July 2020. All patients underwent liver transplantation between November 2006 and April 2020. Clinical and biological data were also recorded for every patient included in the study from their medical files. Abnormal liver enzymes were defined by alanine transaminase > 41 U/L, aspartate transaminase > 40 U/L, alkaline phosphatase > 130 U/L, GGT > 60 U/L or total bilirubin (TBil) >21 μmol/L. The ethics committee of the University Hospital of Montpellier granted ethical approval (Number: 198711) and all patients signed an informed consent.

**Reporting summary**. Further information on research design is available in the Nature Research Reporting Summary linked to this article.

## Data availability

The authors declare that all data supporting the findings of this study are available within the paper, its supplementary information files, or public repositories. NGS data have been deposited in the NCBI SRA database with the BioProject ID: PRJNA714981. The flow cytometry data have been deposited in FlowRepository (https://flowrepository.org), with IDs: FR-FCM-Z3K6 (Fig. 2b), FR-FCM-Z3KL (Fig. 2d), FR-FCM-Z3KM (Fig. 2e), FR-FCM-Z3KT (Fig. 3e_V3_7_11), FR-FCM-Z3KU (Fig. 3e_V14_16_18), FR-FCM-Z3KV (Fig. 3e_V19_22_78), FR-FCM-Z3KR (Fig. S3A), and FR-FCM-Z3LZ (Fig. S3B). Source data are provided with this paper.

## Code availability

Python scripts used to process NGS data are available from github at: https://github.com/hungjuchang/NGS-Sequence-Counts_PWM_PSSM-calculator.git

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

## Acknowledgements

We thank members of the synthetic biology group and the CBS for fruitful discussions. We are grateful to the patients for participating in this study and providing their samples, and to the personnel of the Montpellier CHU hospital for collecting and preparing the samples. This work was supported by an ERC starting grant "COMPUCELL" to J.B. J.B. also acknowledges support from the INSERM Atip-Avenir program and the Bettencourt-Schueller Foundation. The CBS acknowledges support from the French Infrastructure for Integrated Structural Biology (FRISBI) ANR-10-INSB-05-01.

## Author contributions

H.-J.C., A.Z., G.C. and J.B. designed experiments. H.-J.C., A.Z., P.L.V., E.F.-R. performed experiments. H.-J.C., A.Z., P.L.V., E.F.-R., G.C. and J.B. analyzed experiments. I.C., L.M., M.M. and G.-P.P. collected clinical samples and analyzed clinical data. J.G. performed bioinformatic analysis and structural modeling of TcpP. H.-J.C., J.G. and M.C.-G. analyzed the structural data. H.-J.C. and J.B. wrote the paper. All authors participated in elaborating the final version of the manuscript and approved it.

## Competing interests

The authors declare the following competing interests: two patent applications related to this work have been deposited. H.-J.C., J.G., M.C.G. and J.B. are named inventors on Patent application number PCT/EP2021/066224 (filed 16 June 2021), which is related to the TcpP/TcpH receptor system described in this article. H.-J.C. and J.B. are named inventors on the patent application EP21305165.9 (filed 8 Feb 2021), which is related to the VtrA/VtrC receptor system described in this article. The remaining authors declare no competing interests.
