## [Peer Review File · Nature Communications]

Reviewer #1 (Remarks to the Author):

The authors present a revised version of the previously submitted manuscript applying the EMERALD framework to develop a bile acid sensor. The authors remove aspects and sections that were previously suggested as potentially unsupported by their data, ranging from the title's claims to being portable, to correlations that were of uncertain persuasiveness or importance. The work in general seems solid. In short, this reviewer thinks that the paper has improved slightly scientifically, but that the overall big picture remains the same and is unpersuasive for Nature journals but should be an excellent paper in ACS Synthetic Biology.

The response to the previous reviews spends some time arguing that engineering is iterative, and thus publication of their previous work should not preclude publication of this work. Precluding publication of this work is not at all what this reviewer ever said; in fact, this reviewer said this would be a "solid, if not excellent" ACS Synthetic Biology paper! But just not sufficiently impactful or broadly interesting to be in Nature Chemical Biology (or now, in Nature Communications). The authors present some examples that appear to argue that previous second papers have been published in high-impact journals; this may be true, but they don't account for the fact that those papers typically include more novelty than just a second application, or come much closer in development to a finalized application that is point of care or some other valuable characteristic. The authors have now removed their claims to being point of care, and so the remainder of their novelty comes from applying their previously-developed framework to a specific problem and demonstrating it in patient samples, but still with a long way to go before translational application. That is very much what this reviewer would expect from an ACS Synthetic Biology paper, but not a Nature Communications paper. This reviewer has been involved in other reviews where a paper submitted to Nature Communications was deemed to be too similar to previously published work, and those authors added a significant amount of novelty, application, and expansion in response to those critiques. The authors do not really do that here.

The authors present the following justification for their novelty as key advances:

1. We produce a synthetic bacterial receptor programmed to detect a clinically relevant biomarker - to our knowledge this is a first.
2. Here we plug in two other sensing domains - meaning it's scalable.
3. We show a new circuit design with protease control.
4. We demonstrate that directed evolution can be used to improve the receptor.
5. We show a whole-cell bactosensor colorimetric assay, to our knowledge the fastest to date.
6. We suggest potential for other applications, such as basic research on sensing modules and drug screening."

But these don't all ring true. For #1, this claim of novelty/advance/impact rests on the fact that caffeine is not clinically relevant but bile salts are. That's true but a hard sell as a justification for impact, when the remainder of the development doesn't take things much closer to translational application. For #2, they call it scalable, but they had to do protein engineering to get it to work sufficiently. That does not really comport with the common understanding of the meaning of the word "scalable"; applying the same approach in two other applications may demonstrate that it might be "generalizable", but "scalable" means it is able to be done "at scale", and the steps and effort required for the post-fusion engineering really do not suggest scalability. For #3, what they mean by the protease control is unclear; it appears they are talking about a confounding aspect of their specific sensor, so this isn't a generalizable advance. For #4, that is expected from the technique they used, not a unique advance. For #5, there have been fast whole cell sensors before, with some summarized in <https://doi.org/10.1016/j.copbio.2016.11.023> (though not colorimetric), and a colorimetric example in <https://doi.org/10.1021/acssynbio.9b00348> in 60

minutes. And #6 isn't an advance, it's discussion. Actually executing some of the proposed ideas would be more of an advance. The authors have deemed a number of potential advances out of scope for the paper. That is fine, that is totally understandable, we have all said and done that... but a consequence of that may be that the paper's scope is out of scope/impact range for certain journals.

The authors' responses make it seem they think this reviewer says their work is not publishable. That is certainly not the case. It is good stuff. It is read-worthy for people who do biosensors. But is it generally interesting to a Nature Chemical Biology or Nature Communications audience? Is there a fundamental advance, or is it really an application? The latter seems to be most true. It's a good paper... it just is hard to see it as sufficiently broadly interesting and impactful for these journals. Also note that it is not just this reviewer, but also another reviewer, that said the same thing about ACS Synthetic Biology being a good home. One comment could be a cranky reviewer. Two independent comments suggest there is something bigger there.

The authors in their response and in their manuscript do focus on the scalability. It is just not so persuasive to say that anything that involves protein engineering, whether rational or directed evolution or both, is scalable. Protein engineering is not generally referred to as "scalable", otherwise it seems likely we'd have a lot better proteins everywhere around us in biotechnology.

The authors also note in their response that other bile salt sensors measure total bile salts but theirs does only primary bile salts. That may be true, but the challenge of what they are trying to do becomes evident with patients like #17, who have high bile salt levels but who do not get a big response from their sensor. It seems possible that their sensor's profile of what they can detect does not align with what is in that patient, which becomes a potential issue with their non-specific approach to primary bile salts.

There's a lot of continued discussion in the response to reviewers about things from the previous review that may not have a huge impact. Like what is "massively parallel"... this reviewer was not trying to argue about whether saturation mutagenesis or directed evolution is massively parallel, just that the wording can make it seem like the EMeRALD approach and the engineering of many sensors in massive parallel is possible, when it's not. And arguments about whether some things are overstated... like when the authors conclude a response with "To conclude, as of today we don't see any "other method" capable of providing deep mutational exploration, high-throughput screening, and a functional in-cellulo assay to explore sequence-function relationships of bacterial sensing modules." Sure, one can do those things with EMeRALD. Or with a natural transcription factor. Or with any other engineered transcription factor. There is just nothing that is unique to EMeRALD there, that is all this reviewer was saying. A "platform" that includes collectively requiring rational protein evolution, deep sequencing, etc etc, just does not seem to be "scalable" or "rapid" or widely adoptable.

There still seem to be some differences (for example, at 0 GDCA) in Figure S14 for the 30 minute case, but I guess the point is that these are just "representative images" where they selected their preferred ones? If so that seems fine.

The authors address the negative readouts in the response to reviewers but not in the main text. It is only vaguely mentioned in the supplement. This is not sufficient.

So, in conclusion:

The minor concerns mostly seem to be addressed, the questionable scientific conclusions/inferences have been removed, but the bulk of the paper is the same as it was before, besides showing that detection in urine does not work for actual endogenous bile acids.

This reviewer still thinks this would be a great ACS Synthetic Biology paper. But this reviewer is still not convinced that an application of an existing "platform" to a clinical target, with the assay not actually being taken close to translation, would be of broad interest or excitement to a generalist readership like Nature Communications. The fundamental advance one would expect is just not there.

Response to reviewer for NCOMMS-21-21906-T:**Programmable receptors enable bacterial biosensors to detect pathological biomarkers in clinical samples.**

Hung-Ju Chang¹, Ana Zuniga¹, Ismael Conejero^{1,2,3}, Peter L. Voyvodic¹, Jerome Gracy¹, Elena Fajardo-Ruiz¹, Martin Cohen-Gonsaud¹, Guillaume Cambray¹, Georges-Philippe Pageaux⁴, Magdalena Meszaros⁴, Lucy Meunier⁴, and Jerome Bonnet¹.

Affiliations:

¹Centre de Biologie Structurale (CBS). INSERM U1054, CNRS UMR5048, University of Montpellier, France.

²Neuropsychiatry: Epidemiological and Clinical Research, Inserm Unit 1061, Montpellier, France.

³Department of Psychiatry, CHU Nimes, University of Montpellier, Montpellier, France.

⁴Department of Hepatogastroenterology, Hepatology and Liver Transplantation Unit, Saint Eloi Hospital, University of Montpellier, Montpellier, France.

Montpellier, July 27th 2021.

Reviewers' comments are in black, our responses in blue.

Reviewer #1 (Remarks to the Author):

The authors present a revised version of the previously submitted manuscript applying the EMeRALD framework to develop a bile acid sensor. The authors remove aspects and sections that were previously suggested as potentially unsupported by their data, ranging from the title's claims to being portable, to correlations that were of uncertain persuasiveness or importance. The work in general seems solid. In short, this reviewer thinks that the paper has improved slightly scientifically, but that the overall big picture remains the same and is unpersuasive for Nature journals but should be an excellent paper in ACS Synthetic Biology.

The response to the previous reviews spends some time arguing that engineering is iterative, and thus publication of their previous work should not preclude publication of this work. Precluding publication of this work is not at all what this reviewer ever said; in fact, this reviewer said this would be a "solid, if not excellent" ACS Synthetic Biology paper! But just not sufficiently impactful or broadly interesting to be in Nature Chemical Biology (or now, in Nature Communications). The authors present some examples that appear to argue that previous second papers have been published in high-impact journals; this may be true, but they don't account for the fact that those papers typically include more novelty than just a second application, or come much closer in development to a finalized application that is point of care or some other valuable characteristic. The authors have now removed their claims to being point of care, and so the remainder of their novelty comes from applying their previously-developed framework to a specific problem and demonstrating it in patient samples, but still with a long way to go before translational application. That is very much what this reviewer would expect from an ACS Synthetic Biology paper, but not a Nature Communications paper. This reviewer has been involved in other reviews where a paper submitted to Nature Communications was deemed to be too similar to previously published work, and those authors added a significant amount of novelty, application, and expansion in response to those critiques. The authors do not really do that here.

We thank the reviewer for taking the time to evaluate our manuscript in depth. We disagree on some points, and in particular we think the work we show here can appeal to the broad readership of *Nature Communications*.

The authors present the following justification for their novelty as key advances:

- “1. We produce a synthetic bacterial receptor programmed to detect a clinically relevant biomarker - to our knowledge this is a first.
2. Here we plug in two other sensing domains - meaning it's scalable.
3. We show a new circuit design with protease control.
4. We demonstrate that directed evolution can be used to improve the receptor.
5. We show a whole-cell bactosensor colorimetric assay, to our knowledge the fastest to date.
6. We suggest potential for other applications, such as basic research on sensing modules and drug screening.”

But these don't all ring true. For #1, this claim of novelty/advance/impact rests on the fact that caffeine is not clinically relevant but bile salts are. That's true but a hard sell as a justification for impact, when the remainder of the development doesn't take things much closer to translational application.

Again we respectfully disagree on the fact that we don't bring our system closer to translational application. Of course, we already discussed this point in the previous response to referee and corrected the paper to make clear that this is not yet a system usable in the field, but it is still the first example of synthetic receptor platform optimized in whole-cell biosensor and detecting a clinically relevant biomarker at clinical levels, in sample patients. In such, our work might appeal to the broad readership of *Nature Communications*.

For #2, they call it scalable, but they had to do protein engineering to get it to work sufficiently. That does not really comport with the common understanding of the meaning of the word “scalable”; applying the same approach in two other applications may demonstrate that it might be “generalizable”, but “scalable” means it is able to be done “at scale”, and the steps and effort required for the post-fusion engineering really do not suggest scalability.

Here again, we do not agree to switch the reviewer: possibly for what we call scalable, the reviewer has something like “plug-and-play” in mind? To be clear, even the first version of the receptor was working pretty well, potentially to the same sensitivity as in the *V. cholerae* host (although this is hard to assess). Of course, we had to do some tuning: changing expression levels for example, but this is in the range of expected things to do in such systems. Now, about directed evolution, we performed directed evolution to optimize the sensitivity of our system by modifying the sensing domain. In fact we think this a strength of our system. The method we show is efficient, and directed evolution could be used to optimize the behavior of the fusion protein *per se*. We thus think we show that our system can indeed be used at scale, plugging different sensing domains and obtaining various chimeric receptors responding to different ligands. We'd love to have demonstrated more than 2 additional, and are in fact working hard to do this. Regarding the amount of work needed to obtain a working system, it is within or below the current state of the art of sensor engineering.

For #3, what they mean by the protease control is unclear; it appears they are talking about a confounding aspect of their specific sensor, so this isn't a generalizable advance.

We're sorry about confusing the reviewer here. What we meant is that the natural regulatory mechanism we co-opted from the TcpP-TcpH system (continuous proteolysis of the receptor until the ligand is present) might help having a non-leaky and dynamic system and might be repurposed in the future to other sensors.

For #4, that is expected from the technique they used, not a unique advance. There is a difference between expected and demonstrated.

For #5, there have been fast whole cell sensors before, with some summarized in <https://doi.org/10.1016/j.copbio.2016.11.023> (though not colorimetric), and a colorimetric example in <https://doi.org/10.1021/acssynbio.9b00348> in 60 minutes.

We agree that other fast whole cell sensors have been proposed before, with some summarized in <https://doi.org/10.1016/j.copbio.2016.11.023>; Indeed the cell responses can be as fast as in mins (like: DOI 10.1007/s11356-015-6000-7), however the papers summarized here required different machines to support the detection step.

Regarding the colorimetric example, if our understanding is correct, the authors use a colorimetric assay for the cell-free sensor, not the whole cell. In addition, note that our system operates in complex serum samples, notorious to produce interference, and can still produce a fast response.

And #6 isn't an advance, it's discussion. Actually executing some of the proposed ideas would be more of an advance. The authors have deemed a number of potential advances out of scope for the paper. That is fine, that is totally understandable, we have all said and done that... but a consequence of that may be that the paper's scope is out of scope/impact range for certain journals. We agree this has not been demonstrated yet by ourselves, and as we say we *suggest* other applications.

The authors' responses make it seem they think this reviewer says their work is not publishable. That is certainly not the case. It is good stuff. It is read-worthy for people who do biosensors. But is it generally interesting to a Nature Chemical Biology or Nature Communications audience? Is there a fundamental advance, or is it really an application? The latter seems to be most true. It's a good paper... it just is hard to see it as sufficiently broadly interesting and impactful for these journals. Also note that it is not just this reviewer, but also another reviewer, that said the same thing about ACS Synthetic Biology being a good home. One comment could be a cranky reviewer. Two independent comments suggest there is something bigger there.

We don't know what "something bigger" means. As we said, we totally like and respect ACS synthetic biology, in fact our first version of the receptor (without an application) was published there; this work with all the additions that we think it provides (i.e. the difference between a "one-off" example and a platform that could be used for real-world applications) is in our opinion believe are making it of broad interest and so we decided to submit to Nature Journals. We did not want to hurt the reviewer's feelings by making him/her think we did not understand his/her appreciation of our work. We are grateful for the thorough review, the appreciation of the work, the corrections that were made, and this reviewer has made this paper better for sure. It seems the only point we cannot agree on is that the advance provided and the interest for a broad audience, we have already explained our

points. Without any arrogance, we have worked several years on the topic (as the reviewer probably), and we genuinely think that our work will be recognized as a landmark in whole-cell biosensor engineering and applications to the clinics.

The authors in their response and in their manuscript do focus on the scalability. It is just not so persuasive to say that anything that involves protein engineering, whether rational or directed evolution or both, is scalable. Protein engineering is not generally referred to as “scalable”, otherwise it seems likely we’d have a lot better proteins everywhere around us in biotechnology.

Please see response to #2 above. In all, the reviewer comment is a bit too general in our opinion. Saying that anything that uses directed evolution is not scalable is a bit harsh! directed evolution can be used to optimize expression levels, find better sequences for linker regions, change sensitivity, or ligand specificity, etc. This can be made even more efficiently with computational modeling like we did and NGS analysis (also using designer oligo libraries). An important point is to have a workflow that can be accomplished in a few weeks, and we show this here. In the future, we expect the industrial processes will allow even simpler and faster optimization of the receptors. And regarding “better proteins”, we are confident that with the recent progresses in *ab initio* prediction coupled with directed evolution, many are already in the pipeline or coming.

The authors also note in their response that other bile salt sensors measure total bile salts but theirs does only primary bile salts. That may be true, but the challenge of what they are trying to do becomes evident with patients like #17, who have high bile salt levels but who do not get a big response from their sensor. It seems possible that their sensor’s profile of what they can detect does not align with what is in that patient, which becomes a potential issue with their non-specific approach to primary bile salts.

I think the reviewer might want to mention about patient #7 or #16 rather than #17.

The (positive) correlations between primary conjugated bile salts with different liver diseases have been well characterized through LC/MS method by different groups (Trottier et al. 2012; Sugita et al. 2015; Luo et al. 2018; Elsheashaey et al. 2020). Indeed, there might be potential factors such as diets, medical treatments, metabolic issues that may interfere with the performance of bactosensor. Further large scale analysis of the chemical components in clinical samples, and monitoring diet condition/ medical treatments for each patient could help us to understand the limitation of bactosensors.

There’s lot of continued discussion in the response to reviewers about things from the previous review that may not have a huge impact. Like what is “massively parallel”... this reviewer was not trying to argue about whether saturation mutagenesis or directed evolution is massively parallel, just that the wording can make it seem like the EMeRALD approach and the engineering of many sensors in massive parallel is possible, when it’s not. And arguments about whether some things are overstated... like when the authors conclude a response with “To conclude, as of today we don't see any “other method” capable of providing deep mutational exploration, high-throughput screening, and a functional in-cellulo assay to explore sequence-function relationships of bacterial sensing

modules.” Sure, one can do those things with EMeRALD. Or with a natural transcription factor. Or with any other engineered transcription factor. There is just nothing that is unique to EMeRALD there, that is all this reviewer was saying. A “platform” that includes collectively requiring rational protein evolution, deep sequencing, etc etc, just does not seem to be “scalable” or “rapid” or widely adoptable

We tried to answer as precisely as possible on the different points raised. Many of the points discussed above have been already discussed at length in the previous response to the referee and in this one, so we think we have already answered all these points and we will not continue here.

There still seem to be some differences (for example, at 0 GDCA) in Figure S14 for the 30 minute case, but I guess the point is that these are just “representative images” where they selected their preferred ones? If so, that seems fine.

Yes, these are representative images, while the curves represent absorbance measurements on triplicate on three biological replicates.

The authors address the negative readouts in the response to reviewers but not in the main text. It is only vaguely mentioned in the supplement. This is not sufficient.

We added paragraphs about negative readouts in the result and discussion sections of the main text.

So, in conclusion:

The minor concerns mostly seem to be addressed, the questionable scientific conclusions/inferences have been removed, but the bulk of the paper is the same as it was before, besides showing that detection in urine does not work for actual endogenous bile acids.

This reviewer still thinks this would be a great ACS Synthetic Biology paper. But this reviewer is still not convinced that an application of an existing “platform” to a clinical target, with the assay not actually being taken close to translation, would be of broad interest or excitement to a generalist readership like Nature Communications. The fundamental advance one would expect is just not there.

We are sorry the revised version could not convince this reviewer, and we just do not share his/her point of view as we already explained in our response. Nevertheless, we thank the reviewer again for the deep work, discussion, and comments that have in our opinion improved our paper.